# Perceived Pressures and Mental Health of Breastfeeding Mothers: A Qualitative Descriptive Study

**DOI:** 10.3390/healthcare12171794

**Published:** 2024-09-08

**Authors:** Abigail Wheeler, Shanti Farrington, Fay Sweeting, Amy Brown, Andrew Mayers

**Affiliations:** 1Psychology Department, Faculty of Science and Technology, Bournemouth University, Poole BH12 5BB, UK; sfarrington@bournemouth.ac.uk (S.F.); fsweeting@bournemouth.ac.uk (F.S.); andymayers92@gmail.com (A.M.); 2Public Health, School of Health and Social Care, Swansea University, Swansea SA2 8PP, UK; a.e.brown@swansea.ac.uk

**Keywords:** breastfeeding, postnatal care, mental health, social support, qualitative

## Abstract

When a mother is supported to breastfeed, the benefits for her mental health are significant. However, if pressured or unsupported, the opposite is true. This research examines mothers’ breastfeeding experiences, exploring how perceived pressure can impact perinatal mental health. A sample of 501 respondents to a research questionnaire was explored using Reflexive Thematic Analysis. Three main themes identified were perceived pressure to breastfeed, perceived pressure not to breastfeed and mental health impact. The main findings were that mothers received conflicting advice from healthcare professionals, and pressures to feed in a certain way came from their support networks, as well as from their internal beliefs. Perceived pressures negatively impacted maternal mental health, while positive breastfeeding experiences benefitted mental health outcomes.

## 1. Introduction

Breastfeeding can be a dynamic process, for example, mothers can breastfeed until their child is several years of age, as well as tandem feed if they choose and are able to. Recent research shows that tandem feeding is not a common practice in the UK or US. This is in part due to stigma, but also advice from healthcare professionals, which suggests milk quality is lower when tandem feeding. This research showed that tandem feeding an older and younger infant did not adversely affect the quality of breastmilk, and, therefore, healthcare staff advice should reflect this [1].

Despite the World Health Organization promoting breastfeeding for up to 2 years and beyond, only 25% of women in the UK breastfeed post 6 months [2]. In the Portuguese context, mothers also reported being pressured to stop breastfeeding within a socially acceptable timeframe [3]. Similarly, in the UK, mothers experienced a shift in negativity towards their breastfeeding practices when their child transitioned from ‘baby’ to ‘toddler’. This led to a reluctancy in seeking advice from healthcare professionals and breastfeeding publicly [4].

The relationship between breastfeeding and mental health is complex. Breastfeeding can support maternal wellbeing. However, when challenges arise, the experience can be damaging. Evidence suggests that women who cannot breastfeed and use formula experience stigma, which negatively impacts their perinatal mental health [5]. Research shows that women who could not breastfeed experienced internalised guilt and self-stigma, potentially causing consequences for their mental health at this critical time. This was similar for mothers who chose not to breastfeed. Mothers expressed that they felt guilty because they did not breastfeed to prioritise their own wellbeing owing to a lack of support [6].

In the UK, there is a disparity between private and public maternity care, with research highlighting strengths in private care and specific weaknesses in public high-risk care. Mothers were frustrated at conflicting medical advice and reported feelings of frustration after being unknown by their healthcare team, often having to repeat medical history to different providers [7]. A collaborative approach could bridge the gap between the different support services involved in perinatal care and reinforce positive practices and support for breastfeeding mothers.

Exploring the effectiveness of online breastfeeding support groups, Regan and Brown found that a lack of professional support was a key reason for mothers seeking help elsewhere. A key sub-theme was “Lack of Professional Support”. Mothers reported several shortcomings in the treatment they received from healthcare professionals,, the first being that there was a lack of support due to inadequate training or strain on services. Another theme showed that information from healthcare professionals was contradictory, and this led mothers to feel overwhelmed and confused. Mothers felt there was a “breast is best” emphasis, but this was not backed up by any supporting information or advice postnatally [8]. This research highlights a need for more adequate training for health professionals, better communication between professionals and antenatal advice to match postnatal support.

While studies often focus on perceived pressures from healthcare professionals, there is very little on broader breastfeeding experiences. The aim of this research was to explore the experiences of breastfeeding mothers, examining how perceived pressure can impact maternal wellbeing.

## 2. Materials and Methods

### 2.1. Study Design

This study used a qualitative descriptive design aimed at gaining in-depth insights into breastfeeding mothers’ experiences. Mothers completed an online questionnaire between January and February 2021. The data were originally analysed in 2021 for a dissertation paper. The data were later re-analysed in 2023 for the current study.

### 2.2. Setting and Sample

A convenience sample of 926 participants was recruited through social media via Dr Andrew Mayers. The survey was open to mothers over the age of 18 who had given birth to a child prior to the start of the COVID-19 pandemic, regardless of whether they breastfed the infant or not. Participants gave informed consent to take part in the study. Of the 926 participants who responded to the survey, 667 met the inclusion criteria to be analysed as their children were born prior to the COVID-19 pandemic. Participants who gave birth during the COVID-19 pandemic were excluded to ensure results were not skewed by the difference in perinatal care, which was adapted during the pandemic. Removing any incomplete responses and deleting any duplicate data, a final sample of 501 responses was used for the analysis in this study. The years of birth of the children span from 1985 to 2020. While this is a wide date range, which may impact the relevance of the data, it demonstrates a variety of perspectives and allows for a full understanding of breastfeeding experiences over time.

### 2.3. Data Collection

Data were collected between January and February 2021. Use of social media was the primary method for advertising the study for data collection. Breastfeeding organisations were also used to distribute the survey to mothers they supported. Initial screening questions were included to ensure that the potential participants who had their baby during the COVID-19 pandemic, or were under the age of 18, were excluded from the study. Participants were informed that they could withdraw at any time prior to completing the survey.

Participants completed a qualitative online questionnaire created using Qualtrics™ (see Section A.1). The questionnaire was developed with advice and guidance from infant feeding and healthcare organisations/specialists: the Breastfeeding Support Team at Dorset Healthcare NHS Trust, La Leche League, Professor Amy Brown and Dr Wendy Jones. Initially, the questions were proposed by the lead researcher and discussed with the supervisor. They were derived from the literature review, gaps identified and overall research questions. These were then sent to the specialists in the field, Dorset Healthcare NHS Trust and La Lache League for their comments to ensure content validity. Based on their comments, the final questions were developed, as shown in Table 1. Questions focused on mothers’ experiences with perinatal mental health difficulties and the pressures they felt around how they fed their baby, exploring individual feeding experiences. This focus ensured that the questionnaire covered all aspects the research was intended to measure. Demographic information included the age of the mother at the time of birth and year she had the baby. Once completed, participants were debriefed and signposted to organisations that provide support for maternal mental health.

### 2.4. Ethical Considerations

Participants were given the right to withdraw from the study without reason if they felt they could not continue. Participants were signposted towards contacts they could utilise if they felt they became distressed. To address any potential participant distress, the survey questions were developed with help from experts and health professionals in the field to ensure they were worded sensitively.

Participants’ personal data were anonymised from when they submitted the questionnaire. All data were collected and kept in accordance with GDPR and BU regulations. This was explained to participants in the Participant Information Sheet (see Section A.2).

### 2.5. Data Analysis

Reflexive Thematic Analysis (RTA) [9,10] was used to analyse the survey data as it allows the researcher to establish broad themes and sub-themes from a large data set. During Reflexive Thematic Analysis, the subjectivity of the researcher is not viewed as problematic, but, instead, is valued as part of the analytical process. To carry out this Reflexive Thematic Analysis, NVivo 20 (Qualitative Data Analysis Software) was used to analyse the large data set. The lead researcher familiarised herself with the data and generated initial codes, remaining reflexive at each stage (A.W.). Codes were grouped to form initial themes (A.W.), which were then reviewed by the supervisors (A.W., A.M., S.F., F.S.). The themes were named and defined at this stage. The report was then written by the lead researcher (A.W.) and edited with the secondary researchers (A.M., S.F., F.S., A.B.). By utilising Lincoln and Guba’s evaluative criteria of trustworthiness, this research demonstrates credibility, transferability, dependability and confirmability [11]. To mitigate potential biases, the researcher is continually striving to increase her own knowledge and engage with experts and organisations in these fields to ensure that the work is conducted with integrity and sensitivity to the experiences of participants.

### 2.6. Trustworthiness

Trustworthiness in this research can be demonstrated by evaluating the following four criteria: credibility, transferability, dependability and confirmability.

Credibility is demonstrated in this study by the use of diverse participant recruitment sources such as social media. The use of RTA also requires the researcher to continually reflect on their thoughts, decisions and potential biases throughout the research process. This increases the study’s credibility, as the researcher is aware of their potential impact on the data.

Transferability can be seen in the participant descriptions; readers can see the demographics and contexts of the responses and determine the transferability of these to other settings.

Dependability is demonstrated in several ways. One of these is code-recoding, which was carried out by the lead researcher after the initial coding was complete. Some sub-codes were combined and grouped under the main codes. The lead researcher debriefed supervisors and peer researchers regarding the coding throughout the analysis. The data collection methods have been described, RTA was used for data analysis and themes were developed inductively.

Confirmability is demonstrated by the use of reflexivity throughout the analysis process; the researcher reflected on her own positionality and the impact this could have on the interpretation of the data. The use of direct participant quotes ensures that participant voices are accurately portrayed and represented. The lead researcher was supervised by secondary researchers throughout the research process.

#### Author Positionality

RTA acknowledges that researchers bring bias to the analysis. As the lead researcher, I acknowledge my position as a woman who has not personally experienced childbirth and breastfeeding. My interpretation of this research is subjective based on my limited personal experiences, and I recognise that my knowledge and understanding comes from secondary sources such as friends, family, support groups and reviewing research. These sources have enabled me to increase my knowledge on breastfeeding and childbirth and will have influenced this project.

## 3. Results

A total of 501 participants took part in the study, Table 2 shows the location and nature of the births.. Mothers were aged between 17 and 49 at the time of giving birth. A total of 144 women were still breastfeeding at the time of participating in the study. The range in breastfeeding duration was 4 days to 7 years.

From the Reflexive Thematic Analysis, three main themes (and seven sub-themes) were identified: perceived pressure to breastfeed; perceived pressure not to breastfeed; and mental health impact—shown in Figure 1. Themes were developed inductively, and the data were initially coded for overall semantics, then for whether they referred to support, pressure, mental health impact and physical difficulties. For the use in this paper, the pressure and mental health impact codes were the primary focus, and, finally, the specific sub-themes emerged.

### 3.1. Pressure to Breastfeed

This overarching theme shows where mothers felt they were pressured to breastfeed. From 501 total responses, 238 were coded into the themes relating to pressure to breastfeed. This could be pressure from peers, family, healthcare staff or mothers themselves. It also includes instances where mothers felt a sense of guilt or failure when they were unable to breastfeed.

### 3.2. Pressure from Peers

Some women expressed that their peers had put pressure on them to breastfeed their child:

“That and being told from wider family and ‘mum’ groups that i had failed and not a real mum cos I moved to formula”, and “wider family pushed for me to breastfeed” (Participant 289, 34 years old at time of birth in 2017, Planned C-Section, First Child, breastfed for 22 months).

“I felt there was a lot of pressure laid upon me from the moment my child was born to breast feed. I feel for some mothers who choose not to breast feed or indeed cannot, this pressure could have damaging effects” (Participant 371, 21 years old at time of birth in 2010, Vaginal Delivery, First Child, breastfed for 4 weeks).

### 3.3. Pressure from Healthcare Professionals to Breastfeeed

Some felt pressured to only breastfeed by healthcare staff, despite struggling with this:

“The pressure to do so is horrendous and I felt like a failure. My health visitor made me feel rubbish when I told her I had switched to formula” (Participant 273, 31 years old at time of birth in 2017, First Child, Vaginal Delivery, breastfed for 20 months).

“There was a lot of pressure from my midwife and other health professionals before me baby was born to breastfeed” (Participant 404, 30 years old at time of birth in 2019, Vaginal Delivery, First Child, breastfed for 6 months).

### 3.4. Internalised Pressure

Mothers showed that they internalised pressure put upon themselves to breastfeed, and explained the perceived subsequent consequences if they did not succeed:

“Breastfeeding can be an enormous pressure; it can also be lonely and overwhelming. There were times that I dreamed of not being needed quite so much, of running away into the sunset and just looking after myself! Knowing every single feed is down to you is really exhausting” (Participant 103, 30 years old at time of birth in 2019, Planned C-Section, Second Child, breastfed for 14 months).

“No one was putting pressure on me to supplement her diet but I put a lot of pressure on myself to be able to do it right and help her grow” (Participant 193, 26 years old at time of birth in 2019, Vaginal Delivery, First Child, still breastfeeding at 16 months).

Many felt that being the only person able to feed the baby was a huge amount of pressure, leading to feelings of guilt and failure:

“During early weeks it feels like a large burden being the only one able to feed him, and being up all night doing so also had a negative impact on my mental health” (Participant 392, 25 years old at time of birth in 2019, Vaginal Assisted Delivery, First Child, still breastfeeding at 13 months).

“There’s also far too much pressure that breast is best so women guilt themselves and feel like failures if they don’t” (Participant 116, 30 years old at time of birth in 2019, Emergency C-Section Delivery, First Child, breastfed for 14 weeks).

“Feeling like I had failed, that I couldn’t somehow do what others could do. That I’d not given them as good a start. But ultimately, breast feeding and failing for those weeks made me feel soo stressed. I would dread each feed become obsessed with it. Then cry through with the pain. In the end we had to kind of go with happy mum, happy baby, I was so sore and broken that stopping although had a feeling of guilt it made me heal and not feel anxious which helped the baby” (Participant 223, 26 years old at time of birth in 2005, Vaginal Delivery, First Child, breastfed for 2 years).

### 3.5. Pressure Not to Breastfeed

From the data set, 230 responses were identified as negative with regards to breastfeeding. This highlights instances where mothers felt those around them were suggesting they should stop breastfeeding, that they should formula feed or there was general discomfort around breastfeeding.

### 3.6. Pressure from Healthcare Professionals to Formula Feed

Although healthcare professionals offer support during the infant feeding journey, many women felt they were pressured by comments or suggestions made:

“All of my lowest moments in the first year were when health professionals weren’t supportive of breastfeeding. At times it was a battle against them, and that was intensely mentally and emotionally draining. At times it felt like torture, I was convinced my son had Cow Milk Protein Allergy (CMPA) for several weeks before I could outside the paediatric doctors he was, prior to this they had threatened to refer us to social services as they didn’t believe I was feeding my baby (I was spending about 17 h a day feeding my baby or expressing) and had forced us onto formula top ups with the social services threat” (Participant 73, 28 years old at time of birth in 2018, First Child, Vaginal Delivery, still breastfeeding at 2 years and 5 months).

Some expressed that they felt pressure to formula feed or top up when their wishes were to be supported to exclusively breastfeed:

“Neonatal nurses were very unsupportive. Tried to bully me into formular feeding by insisting that all milk volume be measured, telling me I wasn’t allowed to take baby home to breastfeed until I could prove I was pumping certain specified volumes of milk that increased each day, nurses laughed at how little milk I could express” (Participant 464, 32 years old at time of birth in 2018, First Child, C-Section Delivery, still breastfeeding at 2 years and 7 months).

“The support i received in Poole hospital at that time in 2016 before being discharged was terrible/non-existent. The report received from the midwives at home afterwards was also very poor- pressured into formula top ups in the first few days rather than supported to get baby feeding from the breast or even to pump” (Participant 144, 31 years old at time of birth in 2016, First Child, Assisted Vaginal Delivery, breastfed for 15 months).

### 3.7. Pressure from Family

Some mothers had comments made by their family, those who they felt should be their greatest supporters:

“…my childs dad’s side of the family who thought I was weird for breastfeeding” (Participant 5, 25 years old at time of birth in 2017, Vaginal Assisted Delivery, First Child, still breastfeeding at 3 years).

“I couldn’t understand why something so natural was so difficult for me and I couldn’t understand why it felt like the people I loved didn’t want me to keep trying. So I felt like I had to try even harder to prove to myself and to others that I could do it” (Participant 205, 20 years old at time of birth in 2015, Vaginal Delivery, First Child, breastfed for 2 years).

Some mothers perceived that their families felt formula feeding was best, and were seemingly uncomfortable with the concept of breastfeeding:

“Partner wanted to give baby bottles. Partners parents wouldn’t let me b/f baby in their house unless I sat in the bathroom” (Participant 297, 27 years old at time of birth in 2014, Vaginal Delivery, First Child, breastfed for 23 months).

“My nan and aunt and cousins who are my mother figures were very derogatory about feeding, they would tell me to put my boob away, I wasn’t allowed to feed infront of them as they thought it was disgusting” (Participant 408, 30 years old at time of birth in 2018, Emergency C-Section, First Child, breastfed for 6 months).

Some mothers felt that there was pressure to switch to formula feeding when difficulties breastfeeding arose despite wanting to continue breastfeeding:

“…when I struggled my mum and sister would encourage me to formula feed instead and it was hard to get support sometimes as I knew I didn’t want to do this” (Participant 375, 30 years old at time of birth in 2015, First Child, Emergency C-Section, breastfed for 4 years).

Some also experienced support turn into pressure from family:

“My partner and mother would be supportive but at times then pressured me to stop” (Participant 374, 20 years old at time of birth in 2016, First Child, Assisted Vaginal Delivery, breastfed for 4 years).

### 3.8. Discomfort of Others

Some mothers said that they felt other people were uncomfortable or even disgusted at the notion of breastfeeding. This often led to mothers being asked to feed their child out of sight:

“Other family members felt uncomfortable when I fed in front of them. My mother in law actually told me to leave the room to feed when an old uncle was visiting as he would feel uncomfortable” (Participant 198, 32 years old at time of birth in 2015, Vaginal Delivery, First Child, breastfed for 2 years and 6 months).

“Later the difficulties were other people’s attitudes, being asked to breastfeed in the toilet, for example” (Participant 190, 25 years old at time of birth in 1985, Vaginal Delivery, First Child, breastfed for 19 months).

“…others commented and didn’t want me feeding unless I was away somewhere or fully hidden” (Participant 222, 30 years old at time of birth in 2019, Vaginal Delivery, First Child, breastfed for 2 years).

### 3.9. Disapproval at Age of Child

This was the largest sub-theme in this category, with 92 of the 501 mothers expressing that they received disapproval around breastfeeding their child due to the child’s age. Mothers regularly reported their networks being supportive of them breastfeeding, but only until the child was between 6 months and one year old. At this point, the support would turn to negative comments and pressure to wean.

“…once he hit 6 months I started questions about ‘still’ breastfeeding and when would I give him a bottle it would make things easier. At 2.5 years people think it’s weird and say that I should stop as he is too old” (Participant 19, 24 years old at time of birth in 2018, Vaginal Delivery, First Child, still breastfeeding at 2 years).

This notion of a child reaching a socially inappropriate age to breastfeed was also directed at mothers from their family:

“My family was supportive for the first year and uncomfortably tolerated it thereafter” (Participant 69, 29 years old at time of birth in 2017, Vaginal Assisted Delivery, First Child, exclusively breastfed for 12 weeks and then pumped for a further 3 months).

“Family find it strange that I am still feeding, regular comments about ‘boobies are for babies’” (Participant 92, 27 years old at time of birth in 2018, Vaginal Delivery, Second Child, still breastfeeding at 2 years and 6 months).

“Family: ok the first few months, but very much against natural term breastfeeding, feeding an ‘older’ baby or a toddler. I have had frequent comments and looks, and even had a few big rows about this” (Participant 221, 31 years old at time of birth in 2018, Vaginal Delivery, Second Child, still breastfeeding at 30 months).

But many mothers also reported negative comments from healthcare professionals:

“I felt health care professionals were super supportive up to 6 months but after that made me feel like a pariah. Especially so after 18 months. I was particularly knocked when a maternity consultant made me feel stupid for feeding him whilst pregnant with my lockdown baby” (Participant 69, 29 years old at time of birth in 2017, Vaginal Assisted Delivery, First Child, exclusively breastfed for 12 weeks and then pumped for a further 3 months).

“My health visitor being more supportive towards longer term breastfeeding. The WHO states 2 years and beyond. My HV was under the impression that one year was enough” (Participant 105, 24 years old at time of birth in 2015, Vaginal Assisted, Second Child, breastfed for 14 months).

“The Dr was shocked I was still breast feeding at age 2 and asked why?” (Participant 296, 38 years old at time of birth in 2006, Vaginal Delivery, First Child, breastfed for 23 months).

### 3.10. Mental Health Impact

Mothers spoke about the impact that pressure to breastfeed or not to breastfeed, lack of support and birth experience can have on mental health, with 183 mothers sharing experiences of negative impacts on their mental health:

“I knew that if I didn’t overcome the problems, I faced that I would fall into a deep depression. I felt I was fighting for my mental health” (Participant 66, 34 years old at time of birth in 2016, Vaginal Delivery, First Child, breastfed for 12 months).

“Very minimal support in this area. The HV didn’t show for my 6 week check and I was too depressed to chase it. The GP was supportive by the time I decided I needed their help but really the only person that saw and witnessed my decline was my ex partner and he wasn’t trained to deal with that” (Participant 215, 27 years old at time of birth in 2016, Vaginal Assisted Delivery, First Child, breastfed for 2 years).

“It made me feel really low and even when I think about it now (they are 15, 13 + 11) I still feel sad that I somehow wasn’t enough. People would say, oh they’re bottle fed. I felt I was somehow less of a mum” (Participant 223, 26 years old at time of birth in 2006, Vaginal Delivery, First Child, breastfed for 2 years).

On the other hand, for many, breastfeeding was a lifeline to help improve their wellbeing, with 167 mothers reporting breastfeeding having a positive impact on their mental health:

“I suffered significant mental health problems postnatally and breastfeeding helped give me moments of calm and a sense that there was something I was doing right. At 4 months postnatally I developed a hip condition and ended up bed bound, then using a wheelchair and in a lot of pain for months on end. I was not able to look after my children during this time but I was able to carry on breastfeeding—I believe this was a lifeline for my mental health and one of the few reasons I’m still here today” (Participant 164, 40 years old at time of birth in 2019, Vaginal Delivery, Second Child, still breastfeeding at 40 months).

“I am now breastfeeding my second baby and my first experience had giving me a sense of strength and achievement which has been good for my mental health” (Participant 39, 29 years old at time of birth in 2018, Vaginal Delivery, First Child, breastfed for 10 months).

“I felt that breastfeeding was the only things that I could do for him. When struggling with suicidal thoughts, the idea that if I weren’t here he would have to get formula prevented me from acting on these thoughts. Breastfeeding ultimately saved me life” (Participant 339, 27 years old at time of birth in 2017, Assisted Vaginal Delivery, First Child, still breastfeeding at 3 years and 10 months).

## 4. Discussion

The aim of this study was to explore mothers’ perceptions of infant feeding experiences, highlighting difficulties and pressures they felt during their feeding journey and how these have an impact their mental wellbeing. The research demonstrates how mothers often felt pressure, regardless of how they fed their infant. This explains why some women feel there is not enough support to breastfeed, while others feel there is too much pressure to do so—Thomson et al. reported similar findings in their ‘shame if you do, shame if you don’t’ paper [12]. In their study, mothers experienced guilt and shame in relation to how they fed, and this was often internalised, leading to feelings of inadequacy and mothers not seeking support through fear of judgement of others. This study supports their findings and carries considerations for how mothers are supported on their infant feeding journey.

The theme of pressure to stop breastfeeding was highlighted by some mothers who experienced pressure to move to formula feeding earlier than they wanted. While the World Health Organization (WHO) [2] states that breastfeeding is recommended for infants up to 2 years of age and beyond, previous research [3,4] found that women often cease breastfeeding earlier than this due to the pressures they experience from those around them. The current study supports this, as the outcomes indicated that 40% of mothers in our sample breastfed for 13–24 months. The mothers in our study perceived 6 months as being the time they began to receive comments and questions from their family and peers about why they were continuing to breastfeed past this point. They experienced pressures to wean and formula feed, thus demonstrating a pressure to cease breastfeeding and start formula feeding earlier than health guidance suggests.

Mothers expressed that, in some cases, healthcare professionals pressured them to formula feed rather than supporting their wishes to breastfeed. This is supported by findings from Hunt and Thomson, which show that mothers often do not access support for breastfeeding through fear of pressure and judgement from healthcare professionals [13]. This highlights a need for healthcare staff training to better align with WHO guidance. Training should focus on supporting mothers to breastfeed in the first instance rather than pressuring them to bottle or formula feed. Mothers who want to breastfeed long term should be encouraged and supported and given the correct information and guidance to do so. This idea is supported by the findings in McFadden et al.’s study, which highlighted that, when support was given to breastfeed, the duration and exclusivity increased [14].

On the other hand, the outcomes from this study also demonstrated there is pressure to breastfeed. This supports previous research [5,6,15], where it was found that women who cannot breastfeed and therefore formula feed instead experienced guilt and stigma from others as well as internally. Mothers in the current study expressed feelings of guilt or a sense of failure when ceasing breastfeeding earlier than they wanted. Mothers felt immense pressure to breastfeed compared to bottle feed. This pressure came from family members as well as healthcare professionals and an internalised sense of what they should be doing to succeed and be perceived as a good mother. Alternatively, different methods of feeding should be encouraged, depending on each individual circumstance, as this should not be a ‘one size fits all’ practice.

The pressures and judgement mothers received from those around them regarding breastfeeding beyond 6 months were shown to have an impact on their mental wellbeing. Mothers reported feeling pressured to stop breastfeeding and shame for continuing. Mothers expressed that it is important that the benefits of long-term breastfeeding are made common knowledge and for the information to be readily available to enable them to successfully feed in the way they wish to protect their mental health. Dowling and Pontin’s findings support this idea, with many of their participants reporting worrying particularly about publicly breastfeeding an older infant for fear of judgement or comment [16]. Many also talked about the isolation that comes with long-term breastfeeding, feeling set apart from other mothers who had long stopped feeding. Others talked about how once-positive family members later expressed their discomfort with their continued breastfeeding. The isolation they felt led to poorer wellbeing, although some found support groups, which were positive and encouraging.

Many mothers also emphasised the internal pressures they felt to feed in a certain way, and that, if they did not succeed in this, they felt a sense of guilt and failure. They reported this had significant impacts on their mental health. Internalised pressures from the mothers themselves, as well as pressures to feed in a certain way from healthcare professionals, left mothers feeling guilty if they did not conform. This supports previous research, which suggests that internalised guilt and stigma from others can have a significant negative impact on perinatal mental health [6]. This is an important point when it comes to reviewing support systems for mothers. The ‘shame if you do, shame if you don’t’ [12] concept can have a negative impact on mental health, which can lead to poor breastfeeding experiences and a further need for wellbeing interventions.

Some mothers felt that the health visitor went through the motions of checking baby’s weight and measurements but neglected a focus on their mental health. Previous research highlights a lack of continuity of care from midwives and health visitors when it comes to perinatal mental health care. Health visitors, when deciding whether to refer mothers to perinatal mental health services, identified that reduced contact with mothers was a major barrier to identifying their perinatal mental health needs. The study concludes that there is a need for trusts to provide further perinatal mental health training to maternity care staff, as well as a need for further secondary perinatal mental health service provision [17].

Another powerful topic that arose was that successful breastfeeding supported mothers’ wellbeing and gave them a ‘lifeline’ and sense of achievement. This demonstrates the importance of healthcare services supporting mothers to breastfeed successfully in order to support their own wellbeing. When properly supported, many mothers can successfully reach their breastfeeding goals. This can include support from dedicated support groups [14]. Black et al. found that mothers who were part of a Facebook support group had increased self-efficacy and were supported to breastfeed successfully for longer. They highlighted that the normalisation and accessibility of this support increased self-efficacy and that the symbiotic relationship between mothers supported successful breastfeeding. Increased self-efficacy led to increased self-confidence and feelings of empowerment [18]. Support from a mother’s whole network is important to enable successful breastfeeding. Research by Emmot et al. found that successful breastfeeding outcomes are dependent on support from multiple sources. They found that mothers who had “extensive support” from family, friends and healthcare professionals were more likely to breastfeed beyond 2 months compared to those who only had “familial support” or “low support” [19]. Therefore, if a mother has a wide range of support, she can be successful and reach her infant feeding goals.

It is important that mothers receive support to feed their baby, however they are feeding, including mixed and exclusive breastfeeding. There are several reasons why a mother may formula feed her infant rather than breastfeed. These include preference, financial reasons or physical difficulties with breastfeeding. Jackson et al. found, in a systematic review of multiple papers, that guilt and shame were experienced by mothers regardless of the way they fed their infant. For breastfeeding mothers, this guilt came mostly from their family or friends, whereas formula feeding mothers found this came mostly from healthcare professionals and peers. The findings conclude that there is a need for non-judgemental and mother-centred support to minimise guilt for all mothers. They also highlighted a need for emotional support for mothers who are unable to breastfeed despite wanting and trying to do so [20]. The current research supports the idea that guilt is often felt due to pressures from many areas of a mother’s support network, whether that is pressure to breastfeed or not to breastfeed. Either way, the negative impacts on mothers’ mental health can be significant. It is therefore paramount that good-quality support is delivered holistically to enable mothers to achieve their infant feeding goals and thus support their overall wellbeing.

## 5. Strengths and Limitations

The sample size (501 mothers) was large for a qualitative study but was limited to the UK. However, a large data set can pose a higher risk of inconsistency when coding, and reflexivity is a challenge to maintain.

Using a survey can be limiting as the researchers are unable to ask clarifying questions, which would be the case with using an interview method. However, the questionnaire was created using advice and guidance from experts in the field: the Breastfeeding Support Team at Dorset Healthcare NHS Trust, La Lache League, Professor Amy Brown and Dr Wenday Jones. This ensured the questions encouraged responses which reflected the research objective while invoking minimal distress.

RTA as an analysis method allows for a flexible approach with researcher subjectivity directly influencing the results of the study and provides in-depth analysis of participant perspectives. RTA allows for inductive reasoning without pre-existing theories dictating the analysis. However, the researcher’s subjectivity in interpreting these expressions can lead to selective perception or skewed interpretations based on personal biases.

While the researchers strived to demonstrate trustworthiness throughout the research process, it is acknowledged that further steps could be taken to increase this, such as conducting an external audit.

## 6. Conclusions

Overall, findings from this study show that mothers experience pressure to feed their infant in a certain way, often depending on the child’s age. This pressure leads to negative impacts on maternal wellbeing. However, it is also highlighted that successful infant feeding leads to positive mental health impacts. It is hoped that this research will support improving services for all parents, whether they breastfeed or not. The findings highlight a need for research-based data to be actively used when giving mothers advice. Further research should expand on the finding that women feel counselling would be beneficial to their mental health after giving birth, especially following birth trauma.

## Figures and Tables

**Figure 1 healthcare-12-01794-f001:**
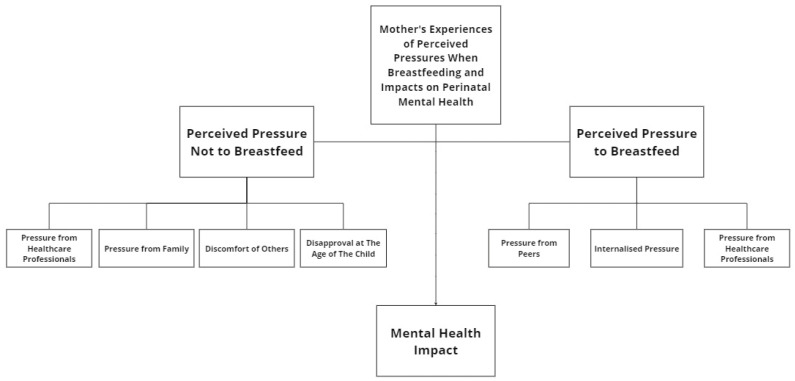
A coding tree showing themes and sub-themes.

**Table 1 healthcare-12-01794-t001:** A table showing the summary of key questions.

Summary of Key Questions
Can you tell us about your breastfeeding experience?
How did those around you feel about you breastfeeding?
Did you encounter any difficulties with breastfeeding?
To what extent did breastfeeding promote your mental health?
What aspects of breastfeeding had a negative impact on your mental health?
Did you receive any support with breastfeeding? If so, from whom?
What further support would you have liked?
What support did you get for your mental wellbeing during this period?
What further support would you have liked for your mental wellbeing?
Is there anything that you feel that would have improved your breastfeeding experience in any way?

**Table 2 healthcare-12-01794-t002:** A table showing participant demographics.

Location of Birth	Number of Mothers
Hospital or Birthing Clinic	456
Home	43
Ambulance	1
Did Not Specify	1
Nature of Birth	
Traumatic	159
Not Traumatic	342
Mother’s Age at Birth	
17–20	11
21–25	55
26–30	178
31–35	178
36–40	69
41+	8
Did Not Specify	2

## Data Availability

The data pertaining to the analysis of this study are available from Abigail Wheeler upon reasonable request.

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
