# Peer review of "Perceived Pressures and Mental Health of Breastfeeding Mothers: A Qualitative Descriptive Study"

_healthcare, 2024, doi:10.3390/healthcare12171794_

Round 1

Reviewer 1 Report

Comments and Suggestions for Authors

Dear Authors,

Thank you for submitting your manuscript to HealthCare Journal. We appreciate the time and effort you have invested in your research. Your study addresses an important topic and provides valuable insights into the experiences and pressures related to breastfeeding and maternal mental health. However, to enhance the quality and impact of your manuscript, we recommend the following revisions:

SUMMARY

Please use MeSH terms for the keywords.

INTRODUCTION

  • There are uncited paragraphs detected. Use current references and high-quality evidence.
  • Very old references with low robustness and scientific quality are detected. Replace these citations with more powerful ones.
  • The research objective has not been explained.

METHODS

  • The design and procedure are not explained. Detail the type of design, the qualitative methodology employed, and the participant selection.
  • Sample: How was the final sample selected? Do you not think that it is too large a sample for a qualitative study on breastfeeding experiences? Do you not think that the sample would have saturated with a smaller number of individuals?
  • Detail when the data collection was conducted and by what procedure.
  • Detail the inclusion and exclusion criteria.
  • Which professionals and experts developed the questionnaire? How many? What was the procedure for developing the questionnaire? Detail this.
  • Did you obtain informed consent from the participants?
  • Clearly state in this section that you selected women with newborns between 1985 and 2020. Breastfeeding and its counseling have evolved significantly over the last 5-10 years. This seems to be a too large a time interval that could lead to confusing conclusions. Why did you choose this time interval? Add some information in the introduction to justify this time interval.

RESULTS

  • In the discussion, you refer to the breastfeeding rates at 24 months. This would be a very interesting data point to reflect in the results, especially if you perform the analysis by years.

DISCUSSION

  • Use more current references and expand the discussion of your work. You discuss your work with only six citations. There is a need to delve deeper into the existing bibliography.

CONCLUSIONS

  • You have not developed conclusions from your research.

OTHER COMMENTS

  • Anonymize the data of the ethics committees (Institutional Review Board Statement).

We believe that addressing these points will significantly improve the manuscript. We look forward to receiving your revised submission and appreciate your cooperation.

Best regards,

Author Response

Dear Reviewer,  

We would like to thank you for your valuable comments.  For the lead author, this is their first experience with peer review, and your comments have been positive and constructive.  We have addressed all your comments in red text and indicated the page number for the amendments. Changes to the manuscript are also in red and blue. I have included a version of the manuscript with the tracked changes accepted.

Thank you again for these comments, we look forward to hearing your thoughts on the updated manuscript. 

Kind Regards 

The Authors

SUMMARY 

Reviewer Comments: Please use MeSH terms for the keywords.  

Our Response:  

  • Thank you, I have edited these to use MeSH terms, see line 25 

INTRODUCTION 

Reviewer Comments: There are uncited paragraphs detected. Use current references and high-quality evidence. 

Very old references with low robustness and scientific quality are detected. Replace these citations with more powerful ones.  

Our Response:  

  • Thank you for pointing these out, I have replaced a couple of older citations with newer ones and re-worded the introduction so that there are no uncited points. 

Reviewer Comments: The research objective has not been explained.  

Our Response:  

  • I have now highlighted the research aims at the end of the introduction section, see line 128. 

METHODS 

Reviewer Comments: The design and procedure are not explained. Detail the type of design, the qualitative methodology employed, and the participant selection. 

Our Response:  

  • I have added a design section to detail the type of design and qualitative methodology used. I have also edited the participant selection section, see line 134 

Reviewer Comments: Sample: How was the final sample selected? Do you not think that it is too large a sample for a qualitative study on breastfeeding experiences? Do you not think that the sample would have saturated with a smaller number of individuals?  

Our Response:  

  • Starting at line 144, I have explained that from the initial 926 responses, those who had their baby during the COVID 19 pandemic were excluded, along with any incomplete results to reach the final sample. 
  • While qualitative research often uses smaller samples than we used, this large sample size captures the complexity and full breadth of breastfeeding experiences. Thematic saturation of core themes could be reached with a smaller sample size, but a large sample shows the full range of experiences, including those less common patterns, which I believe add to the richness of the data and give us a more rounded understanding. The large sample size was an intentional methodological choice to explore of this topic, allowing for depth and breadth to contribute to the broader understanding of breastfeeding experiences, thus fulfilling this study's aim. 

Reviewer Comments: Detail when the data collection was conducted and by what procedure. 

Our Response:  

  • Thank you, I have now added the dates data was collected and by what procedure in the procedure section, see line 162. 

Reviewer Comments: Detail the inclusion and exclusion criteria.  

Our Response:  

  • the “Setting and Sample” section starting on line 143 explains that participants had to be over the age of 18 and not given birth during the pandemic, this was the only inclusion/exclusion criteria 

Reviewer Comments: Which professionals and experts developed the questionnaire? How many? What was the procedure for developing the questionnaire? Detail this.  

Our Response:  

  • The organisations have been blinded for review, but I will detail these in the final manuscript. I have now detailed the input that these organisations had in developing the final questions, see lines 163 and 171 

Reviewer Comments: Did you obtain informed consent from the participants?  

Our Response:  

  • Yes, I have now added this in to make it clear that informed consent was gained from participants, see line 148 

Reviewer Comments: Clearly state in this section that you selected women with newborns between 1985 and 2020. Breastfeeding and its counseling have evolved significantly over the last 5-10 years. This seems to be a too large a time interval that could lead to confusing conclusions. Why did you choose this time interval? Add some information in the introduction to justify this time interval. – 

Our Response:  

  • I accept that this is a large time interval, however, I chose to include responses from as far back as 1985 as the aims of this study are not focused on recent experiences, but rather all experiences. The wide date range demonstrates a variety of perspectives and allows us to understand breastfeeding experiences over time to be able to conclude how better to move forward. Future research using this dataset could break this down to fully appreciate how breastfeeding experiences may have changed over time – I have added a justification on line 156 

RESULTS 

Reviewer Comments: In the discussion, you refer to the breastfeeding rates at 24 months. This would be a very interesting data point to reflect in the results, especially if you perform the analysis by years.  

Our Response:  

  • In this paper we have not analysed the results by years, hence it is not in the results.  
  • However, we do intend to look at this separately in the future. Note I have added a justification on line 156.  

DISCUSSION 

Reviewer Comments: Use more current references and expand the discussion of your work. You discuss your work with only six citations. There is a need to delve deeper into the existing bibliography.  

Our Response:  

  • Thank you for this suggestion, I have now included a couple of more up to date references and also referred to more of the studies mentioned in the existing bibliography  

CONCLUSIONS 

Reviewer Comments: You have not developed conclusions from your research.  

Our Response:  

  •  Thank you for pointing this out, I have now added a Conclusions section 

OTHER COMMENTS 

Reviewer Comments: Anonymize the data of the ethics committees (Institutional Review Board Statement).  

Our Response:  

  • Thank you, I have now anonymised this 

Reviewer Comments: We believe that addressing these points will significantly improve the manuscript. We look forward to receiving your revised submission and appreciate your cooperation. 

Our Response:  

  • Thank you so much for your comments, they have been valuable. 

Reviewer 2 Report

Comments and Suggestions for Authors

An interesting and timely topic.

Introduction needs some polishing, the ideas are there but don’t necessarily flow. Suggest you start with the low rates of breastfeeding, some reasons why it’s challenging to breastfeed including social pressure to do it “right”. Then move into the gaps in the literature and what you plan to do in this study. Work on minimizing the “Author and author found that”.  

Some sentence in the introduction are lengthy and difficult to read – for example “This stigma and pressure from peers and society comes from a lack of understanding and knowledge of the benefits of long-term breastfeeding, these studies conclude that long-term breastfeeding needs to be normalised and greater support and encouragement is needed from health professionals in this area as well as greater representation in the media and other platforms for the benefits of long term breastfeeding.” Consider shortening your sentences to read more smoothly.  

The sentence “This 43 study uses three data collection methods, allowing for a rich data cohort, crucial to the 44 detail expressed in the study.” Seems misplaced

Some word are capitalized that should not be for example Breastfeeding Mothers

Affectively – I think you mean effectively?

The paragraph starting on line 82 is too lengthy, please summarize their findings to what is important to the introduction. I did note that this was the authors prior research and while relevant, it can be summarized succinctly.

Since you acknowledge that researcher subjectivity is inherent in reflexive thematic analysis, could you include a positionality statement for the authors?

In the results there are also sentences that should be split up – for example “The year of birth of each child spans from 1985 to 2020, while this is a wide date range which may impact the relevance of the data,

it is important to note that this does demonstrate the slow progress and lack of vast change of maternity services over this time and was therefore kept in the study and earlier dates 160 not excluded. – suggestyou split between 2020 and while.

My thought with such a wide span is that the internet happened in that time period and that would  likely have made a large impact on the influence of social media.  Did any of the respondents speak to the influence from internet or social media sources?

Line 240 – define CMPA

Limitations section does not include any actual limitations and there are always limitations. For example, survey methodology for qualitative research can be a limiting factor as you cannot ask clarifying questions as you can with interviews or focus groups.

Comments on the Quality of English Language

see above

Author Response

Dear Reviewer,  

We would like to thank you for your valuable comments.  For the lead author, this is their first experience with peer review, and your comments have been positive and constructive.  We have addressed all your comments in red text and indicated the page number for the amendments. Changes to the manuscript are also in red and blue. I have included a version of the manuscript with the tracked changes accepted.

Thank you again for these comments, we look forward to hearing your thoughts on the updated manuscript. 

Kind Regards 

The Authors

Reviewer Comments: Introduction needs some polishing, the ideas are there but don’t necessarily flow. Suggest you start with the low rates of breastfeeding, some reasons why it’s challenging to breastfeed including social pressure to do it “right”. Then move into the gaps in the literature and what you plan to do in this study. Work on minimizing the “Author and author found that”.   

Our Response:  

  • Thank you for this suggestion, I have re-ordered the introduction to follow a similar thread in regard to beginning with the low rates and ending with the gaps 

Reviewer Comments: Some sentence in the introduction are lengthy and difficult to read – for example “This stigma and pressure from peers and society comes from a lack of understanding and knowledge of the benefits of long-term breastfeeding, these studies conclude that long-term breastfeeding needs to be normalised and greater support and encouragement is needed from health professionals in this area as well as greater representation in the media and other platforms for the benefits of long term breastfeeding.” Consider shortening your sentences to read more smoothly.   

Our Response:  

  • I agree, I have reviewed this and made sentences shorter and more concise. 

Reviewer Comments: The sentence “This 43 study uses three data collection methods, allowing for a rich data cohort, crucial to the 44 detail expressed in the study.” Seems misplaced  

Our Response:  

  •  I agree, this has been removed. 

Reviewer Comments: Some words are capitalized that should not be for example Breastfeeding Mothers 

 Our Response:  

  • Thank you for pointing this out, I have now corrected this 

Reviewer CommentsAffectively – I think you mean effectively?  

Our Response:  

  • Thank you for pointing this out, I have now corrected this in line 104  

Reviewer CommentsThe paragraph starting on line 82 is too lengthy, please summarize their findings to what is important to the introduction. I did note that this was the authors prior research and while relevant, it can be summarized succinctly.  

Our Response:  

 - Thank you, I agree and have condensed this paragraph to summarise the key findings, which are important to the introduction  

Reviewer CommentsSince you acknowledge that researcher subjectivity is inherent in reflexive thematic analysis, could you include a positionality statement for the authors?  

Our Response:  

  • Thank you, this is a great idea, I have added a positionality statement to my methods section. See line 254 

Reviewer Comments In the results there are also sentences that should be split up – for example “The year of birth of each child spans from 1985 to 2020, while this is a wide date range which may impact the relevance of the data, it is important to note that this does demonstrate the slow progress and lack of vast change of maternity services over this time and was therefore kept in the study and earlier dates 160 not excluded. – suggest you split between 2020 and while.  

Our Response:  

  • I agree, I have split this sentence as suggested. 

Reviewer CommentsMy thought with such a wide span is that the internet happened in that time period and that would likely have made a large impact on the influence of social media.  Did any of the respondents speak to the influence from internet or social media sources?  

Our Response:  

  • Yes, there were references to the use of online support groups. This research is part of my PhD thesis and given the limited time I have not explored this more in detail. However, in future I would like to revisit this large dataset to further analyse the data and expand on ideas like this. 

Reviewer Comments Line 240 – define CMPA 

Our Response:  

  •  I have expanded on this acronym to show that it refers to Cow Milk Protein Allergy 

Reviewer Comments: Limitations section does not include any actual limitations and there are always limitations. For example, survey methodology for qualitative research can be a limiting factor as you cannot ask clarifying questions as you can with interviews or focus groups.  

Our Response:  

  • Thank you, I agree and have added this, along with other strengths/limitations in this section, see line 623 

Reviewer 3 Report

Comments and Suggestions for Authors

Thank you for the opportunity to review "Mother’s experiences of perceived pressures when breastfeeding and the impact on perinatal mental health" submitted to Healthcare. The study is interesting and relevant to knowledge development in breastfeeding. As I believe the observed limitations in reporting the study can be addressed with a major revision, I took the time to provide detailed feedback and recommendations at the point of observation for improvement in the attached pdf. Please remember the critical but constructive observations are provided to guide a successful  revision that advances the reporting quality of the study in the manuscript. I look forward to reviewing the revised manuscript. 

Summary

To address the study reporting, I ask the authors to review the Standards for reporting qualitative research (SRQR, https://www.equator-network.org/reporting-guidelines/srqr/). Then, please report each element with attention to properly addressing the criteria for each element. Finally, submit the completed checklist indicating the page and paragraph number for each element in the manuscript.

In regard to some elements of the checklist, the audit trail was not established for the methods. The study design was neither stated nor justified. The setting needs to be described and the dates of data collection stated (month/year to month/year). The trustworthiness was superficially addressed in the methods with little evidence about the processes and procedures to substantiate  trustworthiness. The steps for the thematic analysis were addressed in sufficient detail. However, all of these areas can be addressed with revisions, My recommendation is to organize the methods as Study design, Setting and sample, Data collection with sub section structured interview (questionnaire), Data analysis, Trustworthiness, and Ethical considerations. 

The introduction should be consolidated to be clear, concise, and cohesive in presentation with three to five paragraphs ending with the purpose of the study. The section is too wordy, and repetitive in some places. Please make sure this purpose statement is consistent across the sections. There are some minor issues with alignment and cohesion noted in the pdf. The focus of the citations in the introduction are primary for the United Kingdom, which is fine, but please make this focus clear. If needed, move additional important background information for the study to a background section.

The results section is quite dense with quotes. These could be better organized with one or two quotes in each subtheme and the other quotes, and more, presented in a supplemental file. There are some areas for clarification in terms of the meaning of responses versus number of mothers. The discussion section begins strong but gets weaker in terms of supporting references for the discussion. There are only 21 references with some major reviews not cited in the manuscript. The limitations need to be strengthened based upon issues with trustworthiness. There should be a formal conclusion for manuscripts in the health sciences. 

Comments on the Quality of English Language

The language is good and the content is good. There are many wordy and long sentences that need editing to be clearer and more concise in presentation. 

Author Response

Dear Reviewer,  

We would like to thank you for your valuable comments.  For the lead author, this is their first experience with peer review, and your comments have been positive and constructive.  We have addressed all your comments in red text and indicated the page number for the amendments. Changes to the manuscript are also in red and blue. I have included a version of the manuscript with the tracked changes accepted.

Thank you again for these comments, we look forward to hearing your thoughts on the updated manuscript. 

Kind Regards 

The Authors

Reviewer Comments:  To address the study reporting, I ask the authors to review the Standards for reporting qualitative research (SRQR, https://www.equator-network.org/reporting-guidelines/srqr/). Then, please report each element with attention to properly addressing the criteria for each element. Finally, submit the completed checklist indicating the page and paragraph number for each element in the manuscript. In regard to some elements of the checklist, the audit trail was not established for the methods.  

Our Response:  

Thank you for sharing this, I have uploaded the checklist with the requested clarification of page and line number for each element, see attached document

Reviewer Comments:  The study design was neither stated nor justified.  

Our Response:  

I have now addressed this by adding a design section, see line 134 

Reviewer Comments:  The setting needs to be described and the dates of data collection stated (month/year to month/year).  

Our Response:  

Thank you, I have now added this in the “Data Collection” section, see line 162 

Reviewer Comments:  The trustworthiness was superficially addressed in the methods with little evidence about the processes and procedures to substantiate trustworthiness.  

Our Response:  

Thank you, I have now detailed the steps which were taken to address trustworthiness in my research, see line 224 

Reviewer Comments:  The steps for the thematic analysis were addressed in sufficient detail. However, all of these areas can be addressed with revisions, My recommendation is to organize the methods as Study design, Setting and sample, Data collection with sub section structured interview (questionnaire), Data analysis, Trustworthiness, and Ethical considerations. 

Our Response:  

Thank you for this suggestion, I have re-organised the methods section, taking this recommendation into account 

Reviewer Comments:   

The introduction should be consolidated to be clear, concise, and cohesive in presentation with three to five paragraphs ending with the purpose of the study. The section is too wordy, and repetitive in some places. Please make sure this purpose statement is consistent across the sections.  

Our Response:  

Thank you for pointing this out, I have re-worked this section, which is now 5 paragraphs long with a concluding 6th paragraph stating the purpose of the current research 

Reviewer Comments:  There are some minor issues with alignment and cohesion noted in the pdf.  

Our Response:  

Thank you for pointing this out 

Reviewer Comments:   The focus of the citations in the introduction are primary for the United Kingdom, which is fine, but please make this focus clear. If needed, move additional important background information for the study to a background section. –  

Our Response:  

Thank you for pointing this out, it wasn’t entirely intentional, I have added more up to date studies in the introduction, some are not UK based 

Reviewer Comments:   The results section is quite dense with quotes. These could be better organized with one or two quotes in each subtheme and the other quotes, and more, presented in a supplemental file.  

Our Response:  

Thank you for this suggestion, while I agree that the results section is quote heavy, I feel they are warranted in this results section due to the strength they give the data. They are powerful comments from mothers, and each was chosen to highlight the detail and importance of this research. Given the large sample size, it was difficult to limit these quotes to even three per subtheme. 

Reviewer Comments:   There are some areas for clarification in terms of the meaning of responses versus number of mothers.  

Our Response:  

Thank you, I think this may be due to the way I worded these parts, I have now re-worded these to make my point clear, see lines 280 and 340 

Reviewer Comments:   The discussion section begins strong but gets weaker in terms of supporting references for the discussion. There are only 21 references with some major reviews not cited in the manuscript.  

Our Response:  

Thank you for pointing this out, I have now developed the discussion, referring to more of the references used in the existing bibliography 

Reviewer Comments:   The limitations need to be strengthened based upon issues with trustworthiness 

Our Response:  

 I have further developed my limitations section, including issues with trustworthiness, see line 639 

Reviewer Comments:   There should be a formal conclusion for manuscripts in the health sciences.  

Our Response:  

Thank you, I have now added a conclusion section, see line 643 

Regarding the additional comments on the PDF: 

Reviewer Comments: To address the use of purposive sampling  

Our Response:  

the use of breastfeeding specialist organisations ensured that participants met the specific criteria to take part. Social media advertisement also specified the criteria for participants needing to be over 18, given birth to a baby and not during Covid. Therefore, this is purposive and not convenience.  

Reviewer Comments:   To address the large sample size 

Our Response:  

We are justifying the large sample size: While qualitative research often uses smaller samples than we used, this large sample size captures the complexity and full breadth of breastfeeding experiences. Thematic saturation of core themes could be reached with a smaller sample size, but a large sample shows the full range of experiences, including those less common patterns, which I believe add to the richness of the data and give us a more rounded understanding. The large sample size was an intentional methodological choice to explore of this topic, allowing for depth and breadth to contribute to the broader understanding of breastfeeding experiences, thus fulfilling this study's aim. 

Reviewer Comments: To address whether this is a secondary data analysis of an already published study 

Our response: No, it is not a secondary analysis of data from a published study, the data was originally collected as part of an Undergraduate dissertation project and later re-analysed for this paper, hence the additional ethics application 

Round 2

Reviewer 1 Report

Comments and Suggestions for Authors

Good afternoon, thank you for the changes made, however, these are not sufficient for publication. Below are some new suggestions.

  1. Introduction A. You are still using citations with little scientific rigor, such as websites. Please contextualize the topic using scientifically rigorous sources like research articles, not websites. Additionally, I insist on the importance of contextualizing the topic with up-to-date citations. B. The research objective is not clear; it needs to be reformulated. Do not reflect any aspect of the methodology in the objective or introduction (lines 82-83). In this regard, it is much better written at the beginning of the discussion.

  2. Methodology A. In the Study design, do not include any aspects of statistical analysis or sample selection. Reorganize the information. B. When did these women complete the questionnaires? Please describe it. C. It is customary that if subjects do not meet the exclusion criteria, they are not interviewed, not the other way around. Why did you do this? D. The statement in lines 107-108 is false; maternity services have evolved significantly in recent years, for example, with the incorporation of specialists in breastfeeding (such as midwives, IBCLCs, etc.). E. Do not include results in the methodology section. F. Indicate which professionals or organizations helped in the drafting of the questionnaires. The explanation of the validation methodology of these questionnaires needs to be improved as it is ambiguous.

  3. Results A. It would be very interesting if you could add the differences between the experiences of women between 1985-1990 and 2015-2020. I believe there could be interesting similarities and differences that should be highlighted. B. You mention that you do not have data by year. Did you not collect the year within the sociodemographic data? If you do not have this data, it should be reflected as a limitation of the study. C. When introducing a new acronym or abbreviation, first introduce the text and then the abbreviation in parentheses, not the other way around, as seen in line 274.

  4. Discussion A. Limitations: You mention the expert group again. Could you indicate how many and which experts participated in the design of the questionnaires? B. Increase the review of articles in the discussion; you are using websites, the most recent citation is from 2020, and citation 16 is not suitable for a scientific article discussion. Please use current scientific articles and solid bibliography, not gray literature.

  5. Conclusions A. Rewrite the conclusions; they do not respond to the stated objective.

  6. References A. Correct errors in citations 5, 6, 12, 13, 14, 26. B. Remove citations from websites and other gray literature.

Author Response

Dear Reviewer, 

We would like to thank you for your additional comments in this second round of review.  We believe your comments have been constructive and positive in supporting us to ensure this manuscript is the best it can be.  We have addressed all your comments in red text, and we have indicated the page number for the amendments. Changes to the manuscript are also in red. 

Thank you again for these comments, we look forward to hearing your thoughts on the updated manuscript. 

Kind Regards 

The Authors 

Introduction A. You are still using citations with little scientific rigor, such as websites. Please contextualize the topic using scientifically rigorous sources like research articles, not websites. Additionally, I insist on the importance of contextualizing the topic with up-to-date citations.  

Our Response: Thank you for these comments. We have removed the websites, apart from the WHO reference as this comes directly from the WHO website and is necessary to convey the rationale behind this research. All references cited in the introduction are now from within the last 5 years. 

The research objective is not clear; it needs to be reformulated. Do not reflect any aspect of the methodology in the objective or introduction (lines 82-83). In this regard, it is much better written at the beginning of the discussion.

Our Response: Thank you for these comments regarding the introduction. Lines 82-82 have been removed. I have concluded the introduction with a clear objective – see line 103. 

Methodology A. In the Study design, do not include any aspects of statistical analysis or sample selection. Reorganize the information. 

When did these women complete the questionnaires? Please describe it. 

Our Response: Thank you, we have removed aspects of statistical analysis and sample selection. We have added an explanation around when the questionnaires were completed and when data was originally analysed and later re-analysed – see line 113.

It is customary that if subjects do not meet the exclusion criteria, they are not interviewed, not the other way around. Why did you do this? 

Our Response: When participants answered the questionnaire, they were first asked if they had given birth during the COVID 19 pandemic and if they answered “Yes”, they would not have been able to continue and answer the questionnaire. However, upon analysing the data, I found responses where mothers answered “no” but later referenced giving birth during the pandemic – these were therefore removed after data collection to ensure that only responses that met criteria were analysed.  

The statement in lines 107-108 is false; maternity services have evolved significantly in recent years, for example, with the incorporation of specialists in breastfeeding (such as midwives, IBCLCs, etc.). 

Our Response: Thank you, we have removed this statement 

E. Do not include results in the methodology

Our Response: Thank you, I have moved statements including results from the trustworthiness section to the results section. 

Indicate which professionals or organizations helped in the drafting of the questionnaires. The explanation of the validation methodology of these questionnaires needs to be improved as it is ambiguous.

Our Response: Thank you for this comment, we have included the names of professionals/organisations who helped in the development of the questionnaires – see line 149 

We have also developed the methodology of the questionnaire to ensure validity is clear – see lines 152-153 

Results A. It would be very interesting if you could add the differences between the experiences of women between 1985-1990 and 2015-2020. I believe there could be interesting similarities and differences that should be highlighted. 

You mention that you do not have data by year. Did you not collect the year within the sociodemographic data? If you do not have this data, it should be reflected as a limitation of the study. 

Our Response: We agree that this would be an interesting avenue for analysis, however due to the way the data was collected, for the current paper this would not be possible to extract. However, this data set is currently being used for another study and this is certainly something we would like to explore in more depth and report on in the future. 

While we did collect the year of birth for each participant, the primary focus of our analysis was on general experiences rather than changes over time. As such, year-specific trends were not analysed in this study.

When introducing a new acronym or abbreviation, first introduce the text and then the abbreviation in parentheses, not the other way around, as seen in line 274.

Our Response: Thank you for pointing this out, we have now corrected this 

Discussion A. Limitations: You mention the expert group again. Could you indicate how many and which experts participated in the design of the questionnaires?  

Our Response: Thank you for your comment, we have now added who participated in the design of the questionnaire in this section – see lines 580-581

Increase the review of articles in the discussion; you are using websites, the most recent citation is from 2020, and citation 16 is not suitable for a scientific article discussion. Please use current scientific articles and solid bibliography, not gray literature.

Our Response: Thank you, we have removed citation 16 and removed website references (other than the WHO). We have replaced the websites with articles, including recent research from 2021-2023. 

 Conclusions A. Rewrite the conclusions; they do not respond to the stated objective. 

Our Response: Thank you for your comment, we have now re-written the conclusion section to respond to the research objective. 

References A. Correct errors in citations 5, 6, 12, 13, 14, 26. B. Remove citations from websites and other gray literature.  

Our Response: We have removed websites and replaced those and other gray literature with up to date research. We have also reformatted the reference section. 

Reviewer 3 Report

Comments and Suggestions for Authors

Thank you for the opportunity to review the revised manuscript "Mother’s Experiences of Perceived Pressures When Breastfeeding and The Impact on Perinatal Mental Health" submitted to Healthcare." The manuscript has improved with the revisions focused on the methods section. With this stated, the introduction requires revision to present the information relevant to the study in a clear, concise, and integrated manner. Similarly, the abstract needs to be revised to present an overview of the manuscript. Then, the conclusion needs to present the major findings in a clear and concise manner without commentary about the importance of the study. There are some additional observations noted in the manuscript pdf specific to methodological reporting that should be addressed. In particular, please review the information about the sample size in the context of qualitative research and the bias specific to reflexive thematic analysis. There seem to be several misstatements that are indeed general beliefs not supported by the methodological literature. Next, the complete ethics information needs to be inserted into the Institutional Review Board Statement at the end of the manuscript for the data collection of the original study as well as the second study which was a reanalysis. The information is not necessary in the manuscript due to the disclosure section. Finally, the title of the article is not compliant with the reporting guideline for this type of research. There is a suggestion in the attached document as an example. As in the previous review, the observations and feedback are intended to be a constructive critique to improve the presentation of the manuscript. Thanks for the good work to make the first round of revisions!

Comments on the Quality of English Language

The manuscript can use a thorough review by a professional scientific editor as well as a seasoned researcher. There are too many sentences that are vague and areas that need to be stated with clear, concise, and cited sentences. In particular, the abstract, introduction, and conclusion need to be revised for this reason. This observation is not intended to be more than a statement about improving the presentation of the information reported about the study.

Author Response

Dear Reviewer, 

We would like to thank you for your additional comments in this second round of review.  We believe your comments have been constructive and positive in supporting us to ensure this manuscript is the best it can be.  We have addressed all your comments in red text, changes to the manuscript are also in red. 

Thank you again for these comments, we look forward to hearing your thoughts on the updated manuscript. 

Kind Regards 

The Authors 

The introduction requires revision to present the information relevant to the study in a clear, concise, and integrated manner.  

Our Response: Thank you for your comments, the introduction has been revised to be more clear and concise, including more up to date citations also 

Similarly, the abstract needs to be revised to present an overview of the manuscript.  

Our Response: Thank you, I have re-written the abstract to reflect an overview of the manuscript 

Then, the conclusion needs to present the major findings in a clear and concise manner without commentary about the importance of the study. 

Our Response: Thank you for your comment, we have removed commentary regarding the importance of the study and highlighting the major findings, linking to the research objective 

There are some additional observations noted in the manuscript pdf specific to methodological reporting that should be addressed. In particular, please review the information about the sample size in the context of qualitative research and the bias specific to reflexive thematic analysis. There seem to be several misstatements that are indeed general beliefs not supported by the methodological literature.  

Our Response: Thank you for your comment, I have addressed the observations in the manuscript PDF, including those regarding the sample size 

Next, the complete ethics information needs to be inserted into the Institutional Review Board Statement at the end of the manuscript for the data collection of the original study as well as the second study which was a reanalysis. The information is not necessary in the manuscript due to the disclosure section.  

Our Response: This has now been removed from the methods section and inserted into the Institutional Review Board Statement stating both the original study as well as the re-analysis. 

Finally, the title of the article is not compliant with the reporting guideline for this type of research. There is a suggestion in the attached document as an example.  

Our Response: Thank you for your suggestion, we agree with this revision 

To address comments in the pdf 

Reviewer comment: Many of these questions are yes/no rather than open-ended. Was this intentional? 

Our Response: This was not the intention, however these questions had free text boxes and the majority of participants elaborated further than yes/no 

Comments on the Quality of English Language 

The manuscript can use a thorough review by a professional scientific editor as well as a seasoned researcher. There are too many sentences that are vague and areas that need to be stated with clear, concise, and cited sentences. In particular, the abstract, introduction, and conclusion need to be revised for this reason. This observation is not intended to be more than a statement about improving the presentation of the information reported about the study. 

Our Response: Thank you for your comment, the manuscript has been reviewed by seasoned researchers prior to re-submission.